# Developing an implementation blueprint: Lessons learned from integrating electronic patient-reported outcomes in HIV clinics in Alabama

Kelly W. Gagnon[1]*, Kaylee Burgan[1], Morgan Mulrain[1], Stefan Baral[2], Karen Cropsey[3,4], Michael Mugavero[1], Ellen Eaton[1,4]

1 Division of Infectious Diseases, Heersink School of Medicine, University of Alabama at Birmingham, Birmingham, AL, United States of America, 2 Department of Epidemiology, Bloomberg School of Public Health, Johns Hopkins University, Baltimore, MD, United States of America, 3 Department of Psychiatry and Behavioral Neurobiology, Heersink School of Medicine, University of Alabama at Birmingham, Birmingham, AL, United States of America, 4 Center for Addiction and Pain Prevention and Intervention, University of Alabama at Birmingham, Birmingham, AL, United States of America

* kgagnon@uabmc.edu

**Data Availability Statement:** All relevant data are within the manuscript and its Supporting Information Files.

## Abstract

People living with HIV are disproportionately affected by depression, anxiety, and substance use which impede engagement with HIV treatment services and can increase risks of HIV-related morbidity and mortality. Capturing timely, accurate patient data at point of care is recommended to inform clinical decision-making and retain patients on the HIV care continuum. Currently, there is limited use of validated screening tools for substance use and mental health at the point of care in HIV clinics, even though people with HIV (PWH) have a high prevalence of these comorbidities. Even fewer clinics screen in a manner that encourages disclosure of stigmatized substance use, depression, and anxiety. Electronic patient-reported outcomes (ePROs) are an evidence-based modality to overcome such limitations by eliciting responses directly from patients via tablet, smartphone, or computer. To date, there is limited consensus on how to implement ePROs into non-academic settings and enhance uptake. Our team sought to address this gap by examining the implementation of ePROs, previously implemented in an academic clinic, to enhance screening and treatment of mental health (MH) and substance use at five Ryan White HIV/AIDS Program-funded clinics in Alabama. The ePROs were delivered through a multi-component intervention titled HIV + Service delivery and Telemedicine through Effective Patient Reported Outcomes (+STEP), which also provides targeted training to frontline clinicians, and resources for MH and substance use treatment for PWH without access to care. The objective of this study is to provide an implementation blueprint that can be tested and utilized in other HIV clinics to integrate ePROs and increase evidence-based screening for depression, anxiety, and substance use among PWH, as well as outline lessons learned from implementation to date. The findings from this study provide practical steps and advice based on our experience in implementing electronic patient-reported outcomes in HIV clinics in the US Deep South.

**Funding:** This study was funded by the National Institutes of Mental Health (PI: Eaton, 5R01MH124633-04). The funders had no role in study design, data collection and analysis, decision to publish, or preparation of the manuscript.

**Competing interests:** EE reports honorarium from Gilead for participating in the HIV Re-engagement working group and from PRIME, DKBMed, and IAS-USA for developing HIV continuing education content. This does not alter our adherence to PLOS ONE policies on sharing data and materials.

## Introduction

People with HIV (PWH) are disproportionately affected by depression, anxiety, and substance use which impede engagement with HIV treatment services and can increase risks of HIV-related morbidity and mortality [1–5]. Interruptions in HIV treatment and engagement in HIV care can have a negative impact on viral load (VL) suppression among people living with HIV, increasing the risk of HIV progression and transmission [6]. Since poor mental health and substance use are associated with interruptions to HIV treatment, routine screening during point of contact is recommended to identify these co-occurring conditions [7–9]. This is often achieved through the use of validated tools for psychosocial barriers to care such as depression, anxiety, and substance use [6, 10–12]. Routine mental health and substance use screening are recommended by the United States Preventive Services Task Force and the International Antiviral Society- USA for all adults [8, 9].

Despite being recommended for all adults, there is limited routine screening for substance use and mental health conditions beyond depression in HIV clinics, even though PWH have a high prevalence of these comorbidities [13]. Noted barriers to routine substance use and mental health screening are staff hesitation to initiate discussions about these sensitive subjects and limited time during clinical encounters [14, 15]. Electronic patient-reported outcomes (ePROs) are an evidence-based modality to overcome such limitations by eliciting responses directly from patients via tablet, smartphone, or computer [16]. Further, studies have shown that ePROs may mitigate stigma, medical mistrust, and social desirability bias that may impact patients' comfort in reporting symptoms or behaviors as they are completed without a staff member [16, 17]. Lastly, integrating ePROs into routine HIV clinical care provides longitudinal data on symptom severity, care engagement, and patients' psychological and physical well-being [16].

While there are documented processes for establishing ePROs in academic HIV settings, there remains limited consensus on how to implement ePROs into non-academic HIV clinical practice [6, 18]. Given the complexity and barriers to treatment for PWH in non-academic HIV clinical settings, including those in rural areas, our team sought to address this gap by examining the implementation of ePROS to enhance screening and treatment of mental health (MH) and substance use at five Ryan White HIV/AIDS Program (RWHAP)-funded clinics in Alabama (AL) [19]. The ePROs were delivered as part of a multi-component intervention titled HIV + Service delivery and Telemedicine through Effective PROs (+STEP), which also provides targeted training to frontline clinicians, and resources for MH and substance use treatment for PWH without access to care [19].

The objective of this study is to provide an implementation blueprint that can be tested and utilized in other HIV clinics to integrate ePROs and increase evidence-based screening for depression, anxiety, and substance use among PWH. This process, while straightforward in theory, proved quite difficult in practice when implemented in 2022 in under-resourced HIV clinics in the Southeastern US. In addition to sharing the preliminary blueprint and implementation outcomes of +STEP, this study aimed to provide a detailed description of the lessons learned from the implementation of ePROs in five RWHAP-funded clinics across Alabama. Specifically, we identified critical steps and components of integrating and scaling ePROs in HIV clinics that should be accounted for as part of any future efforts to implement ePROs in this clinical setting.

## Materials and methods

The protocol for this study is published in JMIR Research Protocols [19]. In brief, this study utilizes a stepped-wedge hybrid implementation-effectiveness type 1-design to simultaneously

**Table 1. The number of people living with HIV at each Ryan White Funded Clinic.**

| Ryan White Funded Clinics in Alabama | PWH (n) |
|---|---|
| Clinic 1 | 264 |
| Clinic 2 | 2,700 |
| Clinic 3 | 346 |
| Clinic 4 | 550 |
| Clinic 5 | 956 |
| TOTAL (Active PWH at Participating Sites) | 4,816 |

Data from 2022

*Participating site includes a primary clinical site and one more satellite sites

evaluate implementation strategies and the intervention [20]. The +STEP intervention consists of three components: ePROs, targeted staff training, and telemedicine services for MH and substance use. +STEP includes ePROs using validated scales for depression (Patient Health Questionnaire- 9) [21], anxiety (General Anxiety Disorder- 7) [22], alcohol consumption (Alcohol Use Disorders Identification Test-Concise) [23], substance use (Alcohol, Smoking, and Substance Involvement Screening Test) [11], and medication adherence (Self-Rating Adherence Scale-5) [24]. Targeted training is described in detail below. Although ePRO data is collected for the overall study, this manuscript is focused on the preliminary blueprint and lessons learned and does not feature any outcomes from the ePROs individually or in aggregate. The number of PWH receiving care at each of the five +STEP study sites can be found in Table 1. A figure showing the geographic distribution of sites has been previously published [19]. This study was approved by the University of Alabama at Birmingham (UAB) Institutional Review Board and received a waiver of informed consent as data were collected as part of routine clinical care.

After conducting the readiness, acceptability, and accessibility assessment and developing the aforementioned training modules, we began +STEP implementation at Clinics 1 and 2. Due to COVID-19 clinical precautions, implementation began after April 2022. In alignment with our stepped-wedge design, Clinics 3, 4, and 5 launched in October 2022, November 2022, and March 2023, respectively. Throughout the course of implementation, our team developed a preliminary implementation blueprint, a formal implementation plan that describes the specific goals, strategies, personnel, timeframe, and scope involved in the implementation change process [25, 26]. Measures from Proctor's Conceptual Model of Implementation Research were used to assess early implementation outcomes [27]. Specifically, we evaluated adoption of ePROs and effectiveness of clinic workflows on adoption.

To identify lessons learned, our team discussed ongoing challenges and barriers at each step of the implementation process during weekly research team meetings and quarterly Research Team Meetings with experts in HIV and implementation science. The authors also reflected on this process and how issues could be prospectively addressed in future iterations of this work.

## Results

### Preliminary blueprint

We utilized a four-step process to implement +STEP, which serves as the preliminary blueprint to conduct similar work in other HIV clinics (Fig 1). To initiate this study, we identified partner sites by leveraging an existing consortium of all nine RWHAP Part C and D funded

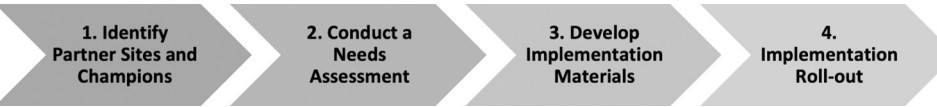

**Fig 1. Implementation blueprint utilized in +STEP.**

clinics across Alabama, the Alabama Quality Management Group (AQMG) [28]. Specifically, we presented the initial concept of +STEP to this consortium in the fall of 2019 and invited all sites to participate. Six sites expressed interest in participation; however, just after funding was granted, one site had to drop out due to competing priorities that arose as a result of COVID-19. We then worked with contacts at each of the five remaining sites to identify site champions. In addition to being primary contacts, champions were responsible for promoting +STEP, training new staff, and troubleshooting issues that arose.

Once sites and champions were introduced to +STEP objectives, we conducted a structured needs assessment with staff, providers, and patients to better ascertain: (1) their current work-flow, (2) their current screening protocols, (3) their telehealth infrastructure, and (4) available resources. Detailed methods and the results of this needs assessment have been previously published [29]. Following the needs assessment, we ranked clinics from "most ready" to "least ready" to implement. We implemented in pairs starting with the clinic that ranked first in our readiness assessment as well as the clinic that ranked fifth, followed by the clinics that ranked second and fourth, and then the clinic that ranked third. This modified stepped-wedge design was utilized to identify barriers, challenges, and confusion surrounding the intervention and its implementation in stages.

The third step in our preliminary blueprint was to develop implementation materials prior to roll-out. Our team began by developing implementation materials that would act as a guide and reference for clinics throughout their launch. Specifically, we developed four asynchronous trainings that would provide targeted education to clinic staff. These trainings covered topics such as MH and substance use stigma, the importance of screening for and treatment of MH and substance use and the pros and cons of telehealth in this space. These trainings were designed for clinic staff, including providers, nurses, social workers, and front desk staff. We asked participating sites to complete trainings prior to their site's launch in return for one hour of Continuing Medical Education (CME) credit and an additional payment installment to their clinic. The research team also compiled an implementation packet, which consisted of (1) contact information for the research team and implementation teams; (2) a pre-implementation checklist; (3) a suggested clinic workflow that details where ePROs may take place and which staff may be responsible for each stage; and (4) screenshots and step-by-step procedures for operating REDCap (REDCap, Vanderbilt University, Nashville, TN) for PRO data entry [30, 31].

Lastly, sites were onboarded to +STEP in alignment with the modified stepped-wedge study design. To onboard, the research team traveled to the sites to meet the implementation teams (champion, front desk staff, clinicians, social workers, case managers) and review onboarding materials. Onboarding materials included a reminder to complete the CME training and confirming that those implementing +STEP had access to REDCap. The research team also checked the iPads to ensure the REDCap shortcut was active and working. After confirming they had access to the REDCap, the implementation teams completed three mock patient encounters to ensure they understood the workflow and asked any questions of the study team. Virtual onboarding sessions were held for any implementation team members who were unable to join in-person meetings. Implementation teams were expected to begin

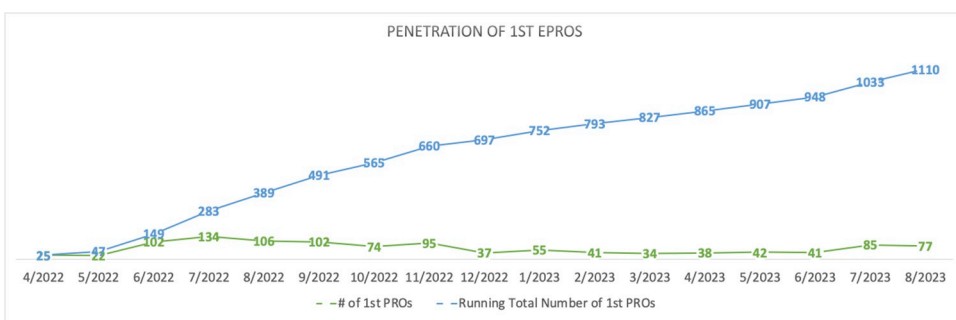

**Fig 2. Number of baseline ePROs each monthly and running total number of baseline ePROs.**

ePROs after being onboarded to the study. A virtual follow-up meeting was scheduled for two weeks after onboarding to discuss progress and identify any challenges. These meetings continued on a monthly basis to ensure the research team could address questions or issues as they arose. The research team maintained open lines of communication with the implementation teams via e-mail between meetings.

## Implementation outcomes

To date, a total of 1,110 baseline ePROs have been completed across sites. As seen in Fig 2, this continuous increase is attributable to an additional 25–135 baseline ePROs each month. This demonstrates a steady roll-out of +STEP across sites and the initial adoption of ePROs among patients who had not yet received an ePRO.

As mentioned in the preliminary blueprint (Fig 1) our study team worked with implementation teams at each site prior to onboarding site visits to discuss where ePROs would be integrated into their workflow, including who was responsible for adding patients to the REDCap and handing them the iPad. In all workflows, patients completed the screeners on the iPads independently, unless they requested assistance. These workflows were modified as needed to align with the staff, resources, and structure of each clinic. Across the five clinics, two main workflows emerged: 1) ePROs are conducted with the patient during their medical visit with mental health staff as the responsible +STEP champion and 2) ePROs are conducted prior to a medical visit with a medical or mental health provider as responsible +STEP champion. A visual of these workflows can be found in Fig 3.

Specifically, two clinics utilize Workflow 1, and three clinics utilize Workflow 2. The effectiveness of these respective workflows is highly dependent on a variety of factors in each clinic, such as staffing and the clinic's patient load. Table 2 presents the number of baseline ePROs collected at each site by workflow utilized. This suggests that Workflow 2 may be more consistent; however, it is worth noting that Clinic 2 had a change in clinic leadership and organizational restructuring that prevented them from participating for six months. The only other clinic currently utilizing Workflow 1 is still in the early stages of implementation and has not had the same time to achieve the level of adoption as other sites.

## Lessons learned

We identified three primary areas of focus that should be considered during any future efforts to implement ePROs in similar clinical settings.

**Clinic engagement.** Before implementing a new intervention, it is critical to understand the clinics you are working with, including their resources, staffing, and primary points of

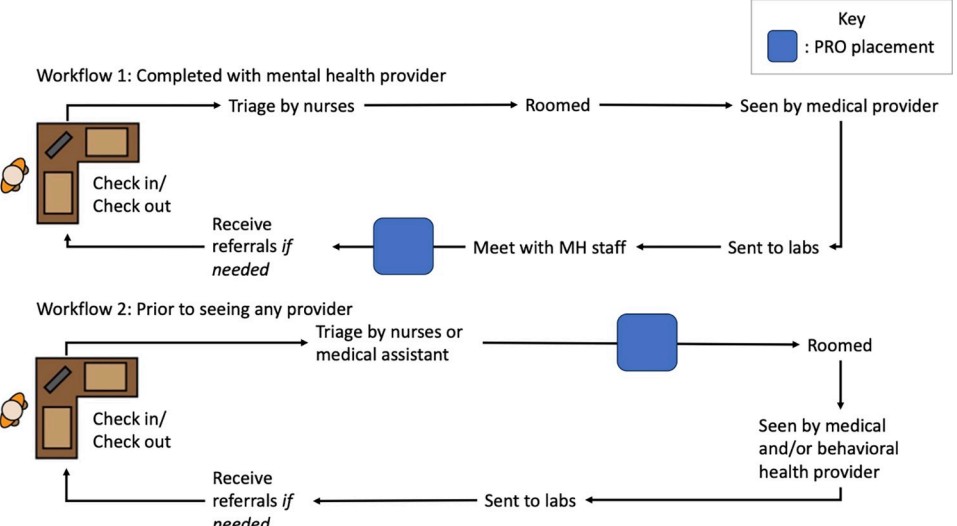

**Fig 3. Workflows implemented by sites to integrate ePROs.**

contact who will help navigate on-the-ground project operations. We found that a productive strategy to initiate engagement was to reach out to existing community groups to present the proposed study and identify points of contact at interested sites. Specifically, during the planning stages of +STEP, our team leveraged AQMG to identify community partners that were willing and interested in implementing +STEP as the new standard of care at their clinic.

During this phase, we also identified a project champion from each site to provide input on +STEP, facilitate communication with other clinic staff, and participate in our needs assessment. These individuals typically held higher-level positions, such as Chief Operating Officer, Chief Executive Officer, or Lead Administrator. Though they could speak to the larger context of their clinics, their expertise was not best suited for understanding on-the-ground clinic operations and perspectives, nor communication of project activities and requirements between staff and the research team. Future studies should enter partnerships with a clearer understanding of which clinical staff could fulfill the role of a project champion, as well as an alternate staff that could fill in as needed.

Additionally, it would have improved our ability to support the sites if we had inquired about the funding landscape of each site, as well as other projects and programs that may cause competition for priorities, resources, and staff engagement. While our needs assessment did evaluate available resources and which staff should be included in the implementation of +STEP, we did not properly address the various priorities and demands associated with other long-term programs that may impede engagement in +STEP. Specifically, it would have been

**Table 2. Workflow utilized by clinic and total number of baseline ePROs to date by clinic\*.**

|  | Clinic 1 | Clinic 2 | Clinic 3 | Clinic 4 | Clinic 5 |
|---|---|---|---|---|---|
| **Workflow 1** |  | X |  |  | X |
| **Workflow 2** | X |  | X | X |  |
| **Total Number of 1st ePROs to date** | 260 | 531 | 144 | 117 | 58 |
| **Proportion of baseline ePROs to Patient Population** | 98% | 20% | 42% | 21% | 6% |

\*Clinics are numbered according to date of onboarding with Clinic 1 implementing the intervention first and Clinic 5 being last.

useful to better understand how many of these clinics were participating in research and programmatic grants. Each of our participating sites had a variety of sponsored projects with UAB, state, and federal funding agencies during the implementation of +STEP. While unknown to our research team, managing multiple studies and funded programs at once created frustrations and confusion regarding the differences between projects, including their goals, expected deliverables, and timelines. This was especially the case for our smaller clinics, which have limited staff capacity. For these sites in particular, the same staff were expected to fill multiple roles across projects resulting in confusion and competing priorities. This highlights how an intervention may be relatively easily adopted in an academic HIV clinic; whereas, implementation may be more challenging at understaffed clinics with limited capacity.

Another area of clinic engagement that arose as a lesson learned was the value of site visits. It was evident that site visits were critical to the onboarding and implementation process. Thus far, we have conducted two site visits to every site, one to launch the intervention (between April 2022 and March 2023) and second to check in, meet new staff, and discuss barriers to the intervention between May and August 2023. From an objective standpoint, we observed an uptick in the number of ePROs directly following site visits; however, we also noted subjective evidence of the value of these visits. Specifically, meeting in person helped the research and implementation teams build rapport and differentiate responsibilities across both groups. Because of site visits, it was clear that implementation teams had a better understanding of who to request support from as issues arose. Since these visits were longer than the monthly check-ins, the research team was also able to elicit more barriers to implementation and collect implementation outcomes data for continuous evaluation.

Further, the research team observed that these visits were often the only time all members of the implementation teams gathered to discuss the progress of conducting ePROs at their site. Thus, these meetings were internally beneficial to improve communication, both within the site and with the research team. Future studies should consider integrating more regular site visits in their protocol to help maintain rapport and relationships with partners and facilitate open communication. Research staff should budget time and resources to allow frequent site visits.

**Incentives.**   Understanding a clinic's motivations for participation is imperative to the success of any implementation study. Prior to implementation, it is necessary to have clear and established guidelines for the delivery of incentives and what benchmarks or objectives need to be met in order for those incentives to be released. Incentives are necessary in order to ensure engagement in ongoing research activities, as well as to provide proper compensation for the time and energy dedicated to the work.

While funding for equipment and ongoing participation was provided to each clinic for their involvement in +STEP, we discovered that these clinics are involved in multiple research studies hosted by UAB. This complicated participation as incentive checks were distributed in bulk from all UAB projects and sites could not differentiate how much they were specifically receiving from +STEP. Further, incentives were dispensed to clinical leadership and not directly to staff implementing +STEP. While it was important to the study team to support the autonomy of the clinics in determining how money was allocated to research tasks, we observed that this incentive model did not act as a form of positive reinforcement for participation. Since staff did not personally receive incentives, there was a critical disconnect between participation and compensation. In future work, we plan to implement a dual incentive model to compensate both the clinic and staff to mitigate this issue.

**Communication.**   At the center of our lessons learned, we identified the importance of early, consistent communication with our study sites and research experts. This includes

ensuring that appropriate representatives are identified, engaged, and at the table during discussions, ongoing review and modifications to implementation materials, and consideration for the dynamics of virtual and physical meeting spaces.

Primarily, it is critical for research team members, regardless of role on the project, to have training and/or experience with communicating with community organizations. A +STEP community liaison, who had established relationships and experience with these organizations, was engaged as part of the research team in an effort to avoid paternalistic behaviors from the research team. These organizations are the experts on their workflow and the needs of their patients, so utilizing an inquisitive approach rather than an instructional one was more productive in evaluating and addressing challenges to implementation.

Prior to visiting the sites for onboarding, the research team attempted to collect the names and roles of staff at the site who would be implementing +STEP at any point during the study period. We found that this was a difficult task, as sites did not fully understand the scope and expectations of the project. In addition, there was notable staff turnover such that staff who needed training evolved over the course of the study. We often collected additional names and contact information during site visits and conducted virtual onboarding for those staff members at a later date. It would have been beneficial to initiate the process with a virtual overview of the project followed by an inquiry as to what staff members would be involved.

Two communication-specific issues arose as it pertains to study materials: 1) specificity of implementation materials and 2) training needs and support. These issues were pronounced as study sites experienced staff turnover and new implementation team members joined the project. The development and use of implementation materials were vital for documenting the purpose and protocols for implementation of +STEP at each clinic; however, there were several opportunities for improvement to make these materials more useful to clinic staff. While our team developed materials that seemed most relevant to each clinic, it was apparent that more detail and nuance would have been beneficial to help guide staff through implementation. In addition to engaging an implementation expert to develop these materials, it was necessary to make ongoing modifications to the implementation materials and workflow documentation. This was of particular importance as new staff joined the teams to facilitate autonomy of each study site in managing both new staff and the implementation process.

As noted under clinic engagement, site visits were essential to implementation, however gathering space at sites was very limited and clinic schedules and patient demands served as barriers to engagement. This required critical evaluation of who needed to be present during site visits, both from the UAB +STEP research team and the sites' implementation teams.

## Discussion

To address the limited consensus on the implementation of ePROs as part of routine HIV clinical practice in non-academic settings, +STEP seeks to establish a blueprint to integrate ePROs into five RWHAP clinics in AL. While this initiative is ongoing, findings from the first year of implementation demonstrate preliminary indicators of a successful blueprint. Specifically, we observed a relatively consistent rate of new ePRO participants over the course of the first year of implementation, accumulating to over one thousand enrolled participants. In addition to sharing the preliminary blueprint, this study reported important lessons learned during the implementation process. The findings from this study serve to begin addressing the literature gap pertaining to the implementation of ePROs to assess mental health and substance use within the context of HIV clinics.

To date, every site in +STEP has onboarded and initiated ePROs with their patients. Participating sites implemented one of two workflows, with initial indications that it is more effective

to conduct ePROs prior to rooming the patient. Previous research focused on process guidance, such as choosing modalities for administering the ePROs and reporting [32–34]; however, these workflows provide a more granular understanding of how ePROs were integrated into HIV clinical care. The two workflow models reported in this study and the corresponding early uptake of ePROs at each site suggest that integrating ePROs prior to rooming the patient may be most achievable for HIV clinics. This corresponds with when screeners, regardless of modality, are administered in most clinical settings [35]. Previous literature suggests this may be because that is when patients have the most independent time and that it is difficult to intervene during other points in the visit, resulting in patients ultimately not completing the screeners [36].

Our preliminary blueprint includes four key steps to initiating and implementing ePROs into routine HIV care. Specifically, the blueprint is a sequential process that builds on itself to achieve implementation. Our four-step blueprint is similar to existing models for implementing ePROs [36, 37], which outline the importance of setting goals for collecting ePROs in clinical practice, understanding the existing context prior to implementation, and training implementation staff. Our preliminary blueprint differs in that we describe the process of identifying partners and leveraging existing community coalitions for intervention and the importance of developing implementation materials to inform the process at each site. As +STEP is an ongoing study, we anticipate that the preliminary blueprint may be modified, including adding further steps as we advance into later implementation and maintenance. Based on existing ePRO frameworks [38], we anticipate steps may include ongoing evaluation and process improvement.

Notably, we identified valuable lessons we learned over course of this process that would be beneficial to informing ongoing and future efforts to integrate ePROs into routine HIV care. Specifically, we shared lessons in three primary areas: clinic engagement, incentives, and communication. Within the context of clinic engagement, input and buy-in from leadership was vital; however, speaking directly with clinic staff and identifying an implementation champion during the first step of the blueprint would have helped us avoid later communication issues. While we did incentivize the clinics, it was a missed opportunity during the needs assessment phase to understand the scope and origin of each clinics' compensation and how that may compete with our initiative. The presence of a community liaison on +STEP facilitated conversations and was overall a notable facilitator to interactions between the study team and clinics. Lastly, implementation materials were critical for each HIV clinic, but we identified the need to maintain these materials as living documents to facilitate implementation, particularly as turnover occurred at sites.

## Conclusions

Our preliminary blueprint aligns with existing models for implementing novel interventions into routine clinical care. These efforts are the first to our knowledge that specify practices for implementing ePROs at RWHAP HIV clinics, particularly in the Deep South. This blueprint demonstrated early usefulness in facilitating implementation through the uptake of ePROs to date. Our lessons learned reflect effective strategies and areas for improvement to inform future efforts to introduce ePROs in HIV clinics. The practical information generated from +STEP thus far advances the field through early insight into the implementation process, outcomes, and lessons.

## Supporting information

**S1 Data. Manuscript data.** This table contains all the data presented in the manuscript in alignment with the data availability statement.
(XLSX)

## Author Contributions

**Conceptualization:** Stefan Baral, Karen Cropsey, Michael Mugavero, Ellen Eaton.

**Data curation:** Kelly W. Gagnon, Kaylee Burgan, Morgan Mulrain, Ellen Eaton.

**Formal analysis:** Kelly W. Gagnon, Kaylee Burgan.

**Funding acquisition:** Ellen Eaton.

**Investigation:** Ellen Eaton.

**Methodology:** Kaylee Burgan, Ellen Eaton.

**Project administration:** Morgan Mulrain, Ellen Eaton.

**Supervision:** Stefan Baral, Karen Cropsey, Michael Mugavero.

**Visualization:** Kelly W. Gagnon, Kaylee Burgan, Karen Cropsey.

**Writing – original draft:** Kelly W. Gagnon, Kaylee Burgan, Ellen Eaton.

**Writing – review & editing:** Kelly W. Gagnon, Kaylee Burgan, Morgan Mulrain, Stefan Baral, Karen Cropsey, Michael Mugavero, Ellen Eaton.

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
