## [Decision Letter · Decision Letter 0]

26 Aug 2024

PONE-D-24-13780Developing an Implementation Blueprint: Lessons Learned from Integrating Electronic Patient-Reported Outcomes in HIV Clinics in AlabamaPLOS ONE

Dear Dr. Gagnon,

Thank you for submitting your manuscript to PLOS ONE. After careful consideration, we feel that it has merit but does not fully meet PLOS ONE’s publication criteria as it currently stands. Therefore, we invite you to submit a revised version of the manuscript that addresses the points raised during the review process. Please address all concerns raised by the reviewers (see below) and in addition, ensure that references and tables adhere to plosone publication requirements. Please submit your revised manuscript by Oct 10 2024 11:59PM. If you will need more time than this to complete your revisions, please reply to this message or contact the journal office at plosone@plos.org. Please include the following items when submitting your revised manuscript:A rebuttal letter that responds to each point raised by the academic editor and reviewer(s). You should upload this letter as a separate file labeled 'Response to Reviewers'.A marked-up copy of your manuscript that highlights changes made to the original version. You should upload this as a separate file labeled 'Revised Manuscript with Track Changes'.An unmarked version of your revised paper without tracked changes. You should upload this as a separate file labeled 'Manuscript'.We look forward to receiving your revised manuscript.

Kind regards,

Ibrahim Jahun, MD, MSc, PhD

Academic Editor

PLOS ONE

Journal Requirements:

When submitting your revision, we need you to address these additional requirements. 1. Please ensure that your manuscript meets PLOS ONE's style requirements, including those for file naming. The PLOS ONE style templates can be found at https://journals.plos.org/plosone/s/file?id=wjVg/PLOSOne_formatting_sample_main_body.pdf and https://journals.plos.org/plosone/s/file?id=ba62/PLOSOne_formatting_sample_title_authors_affiliations.pdf 2. Thank you for stating the following financial disclosure: "This study was funded by the National Institutes of Mental Health (PI: Eaton, 5R01MH124633-04 )." Please state what role the funders took in the study.  If the funders had no role, please state: ""The funders had no role in study design, data collection and analysis, decision to publish, or preparation of the manuscript."" If this statement is not correct you must amend it as needed. Please include this amended Role of Funder statement in your cover letter; we will change the online submission form on your behalf. 3. Thank you for stating the following in the Competing Interests section: "E reports honorarium from Gilead for participating in the HIV Re-engagement working group and from PRIME, DKBMed, and IAS-USA for developing HIV continuing education content." Please confirm that this does not alter your adherence to all PLOS ONE policies on sharing data and materials, by including the following statement: ""This does not alter our adherence to  PLOS ONE policies on sharing data and materials.” (as detailed online in our guide for authors http://journals.plos.org/plosone/s/competing-interests).  If there are restrictions on sharing of data and/or materials, please state these. Please note that we cannot proceed with consideration of your article until this information has been declared.  Please include your updated Competing Interests statement in your cover letter; we will change the online submission form on your behalf. 4. We note that your Data Availability Statement is currently as follows: All relevant data are within the manuscript and its Supporting Information Files. Please confirm at this time whether or not your submission contains all raw data required to replicate the results of your study. Authors must share the “minimal data set” for their submission. PLOS defines the minimal data set to consist of the data required to replicate all study findings reported in the article, as well as related metadata and methods (https://journals.plos.org/plosone/s/data-availability#loc-minimal-data-set-definition). For example, authors should submit the following data: - The values behind the means, standard deviations and other measures reported;- The values used to build graphs;- The points extracted from images for analysis. Authors do not need to submit their entire data set if only a portion of the data was used in the reported study. If your submission does not contain these data, please either upload them as Supporting Information files or deposit them to a stable, public repository and provide us with the relevant URLs, DOIs, or accession numbers. For a list of recommended repositories, please see https://journals.plos.org/plosone/s/recommended-repositories. If there are ethical or legal restrictions on sharing a de-identified data set, please explain them in detail (e.g., data contain potentially sensitive information, data are owned by a third-party organization, etc.) and who has imposed them (e.g., an ethics committee). Please also provide contact information for a data access committee, ethics committee, or other institutional body to which data requests may be sent. If data are owned by a third party, please indicate how others may request data access. 5. Please include captions for your Supporting Information files at the end of your manuscript, and update any in-text citations to match accordingly. Please see our Supporting Information guidelines for more information: http://journals.plos.org/plosone/s/supporting-information.

Reviewers' comments:

Reviewer's Responses to Questions

**Comments to the Author**

1. Is the manuscript technically sound, and do the data support the conclusions?

Reviewer #1: Yes

Reviewer #2: Yes

Reviewer #3: Yes

2. Has the statistical analysis been performed appropriately and rigorously? 

Reviewer #1: Yes

Reviewer #2: N/A

Reviewer #3: N/A

3. Have the authors made all data underlying the findings in their manuscript fully available?

Reviewer #1: Yes

Reviewer #2: Yes

Reviewer #3: No

4. Is the manuscript presented in an intelligible fashion and written in standard English?

Reviewer #1: Yes

Reviewer #2: Yes

Reviewer #3: Yes

5. Review Comments to the Author

Reviewer #1: The manuscript is technically sound and pertinent to improving the quality of care of Person with HIV/AIDS (PWH) because I had previously worked with the author in trying to troubleshoot the best way of improving the health outcome of PWH. The author through this paper has highlighted the critical importance of introducing ePROs to specifically improve the health outcome of PWH particularly those susceptible to poor mental health and substance use.

Reviewer #2: The paper was well written, lesson learnt, and discussion were well aligned. However, the authors need to pay more attention on the use abbreviations. for examples: 1. RWHAP was sometimes written as RHWAP (page 6, 15), this needs to be corrected. 2. SUD - described as substance use instead of substance use disorders. 3. UAB was not described in the paper.

Reviewer #3: 1.Materials and Methods - Lines 103-104 speak of Qualitative Interviews and Structured Surveys. Please provide the respective interview guide and the survey questionnaire. They were not available under the data availability section.

2. Line 214, it will help if the authors could replace "Lessoned Learned" with RESULTS. They can include lessons learnt as a subtitle under RESULTS section.

3. An indication of many site visits was done, will be quite helpful as the authors recommended that there is a need to increase the site visits. So a measurement is needed to show from where this improvement ought to be done.

4. Discussion - It is advisable that the writers restrict their Discussions only to the study findings. They can only discuss what is stated in the results section, except where they are citing other literature in case of findings corroborations.

5. The definition of a Champion should come early, at the first instance of the word champion.

6. References: Please check all your references to ensure they are consistent with the Vancouver style.

6. PLOS authors have the option to publish the peer review history of their article (what does this mean?). If published, this will include your full peer review and any attached files.

Reviewer #1: **Yes: **Emeka Chrisian Madubuko

Reviewer #2: **Yes: **Mukhtar Liman Ahmed

Reviewer #3: **Yes: **Dr. Benson Ncube

---

## [Author Response · Author response to Decision Letter 0]

10 Oct 2024

Response to Reviewers

We sincerely thank the reviewer for their thoughtful and constructive suggestions, which have substantially strengthened our manuscript. Responses to the reviewer’s points are below. The reviewer’s original comments are listed followed by our response. Our responses correspond to changes highlighted in the revised article text and titled ‘Revised Manuscript with Track Changes'. A clean version has also been included, titled 'Manuscript'.

EDITOR COMMENTS TO AUTHOR

We have updated the manuscript to meet PLOS ONE’s style requirements.

"This study was funded by the National Institutes of Mental Health (PI: Eaton, 5R01MH124633-04 )."

"E reports honorarium from Gilead for participating in the HIV Re-engagement working group and from PRIME, DKBMed, and IAS-USA for developing HIV continuing education content."

4. We note that your Data Availability Statement is currently as follows: All relevant data are within the manuscript and its Supporting Information Files.

We have updated the data availability statement to reflect the added data file. The updated data availability statement is in the cover letter and we have uploaded the data file as Supporting Information.

We have added the Supporting Information files to the end of the manuscript. 

We have reviewed our reference list to ensure that it is complete, correct, and does not contain any retracted papers. 

REVIEWER 1 COMMENTS TO AUTHOR

The manuscript is technically sound and pertinent to improving the quality of care of Person with HIV/AIDS (PWH) because I had previously worked with the author in trying to troubleshoot the best way of improving the health outcome of PWH. The author through this paper has highlighted the critical importance of introducing ePROs to specifically improve the health outcome of PWH particularly those susceptible to poor mental health and substance use.

We appreciate your comment.

REVIEWER 2 COMMENTS TO AUTHOR

The paper was well written, lesson learnt, and discussion were well aligned. However, the authors need to pay more attention on the use abbreviations. for examples: 1. RWHAP was sometimes written as RHWAP (page 6, 15), this needs to be corrected. 2. SUD - described as substance use instead of substance use disorders. 3. UAB was not described in the paper.

Thank you for your comment! We have corrected these abbreviations throughout.

REVIEWER 3 COMMENTS TO AUTHOR

1.Materials and Methods - Lines 103-104 speak of Qualitative Interviews and Structured Surveys. Please provide the respective interview guide and the survey questionnaire. They were not available under the data availability section.

Thank you for this comment. Lines 103-104 were intended to describe the overall intervention and not the methods of this manuscript specifically. However, we agree that this is confusing to the read so have removed the aims of the larger intervention from the methods section.

2. Line 214, it will help if the authors could replace "Lessoned Learned" with RESULTS. They can include lessons learnt as a subtitle under RESULTS section.

We have made this correction.

3. An indication of many site visits was done, will be quite helpful as the authors recommended that there is a need to increase the site visits. So a measurement is needed to show from where this improvement ought to be done.

Thank you for this comment. We have added this information on lines 271 to 274.

“Thus far, we have conducted two site visits to every site, one to launch the intervention (between April 2022 and March 2023) and second to check in, meet new staff, and discuss barriers to the intervention between May and August 2023.”

4. Discussion - It is advisable that the writers restrict their Discussions only to the study findings. They can only discuss what is stated in the results section, except where they are citing other literature in case of findings corroborations.

We agree with the reviewer’s sentiment regarding the scope of discussions but were not able to identify anywhere in our discussion that goes outside of the scope of our results or cited literature. 

5. The definition of a Champion should come early, at the first instance of the word champion.

Thank you for this suggestion. We removed the initial first mention of champions so that the definition follows the first mention of champion on lines 159-161. 

6. References: Please check all your references to ensure they are consistent with the Vancouver style.

Thank you again for the careful assessment. We have further reviewed the references and corrected the journal names to the abbreviated titles.

---

## [Editor Report · Decision Letter 1]

14 Oct 2024

Developing an Implementation Blueprint: Lessons Learned from Integrating Electronic Patient-Reported Outcomes in HIV Clinics in Alabama

PONE-D-24-13780R1

Dear Dr. Gagnon,

We’re pleased to inform you that your manuscript has been judged scientifically suitable for publication and will be formally accepted for publication once it meets all outstanding technical requirements.

Kind regards,

Ibrahim Jahun, MD, MSC, PhD

Academic Editor

PLOS ONE
---

## [Editor Report · Acceptance letter]

18 Oct 2024

PONE-D-24-13780R1 

PLOS ONE

Dear Dr. Gagnon, 

I'm pleased to inform you that your manuscript has been deemed suitable for publication in PLOS ONE. Congratulations! Your manuscript is now being handed over to our production team.

Kind regards, 

on behalf of

Dr. Ibrahim Jahun 

Academic Editor

PLOS ONE